# Bad Seeing or Bad Thinking?
# Rewarding Perception for Multimodal Reasoning

**Haozhe Wang**[1]  **Qixin Xu**[2]  **Changpeng Wang**[3]  **Taofeng Xue**[4]  **Chong Peng**[4]  **Wenhu Chen**[5]  **Fangzhen Lin**[1]

## Abstract

Achieving robust perception-reasoning synergy is a central goal for advanced Vision-Language Models (VLMs). Recent advancements have pursued this goal via architectural designs or agentic workflows. However, these approaches are often limited by static textual reasoning or complicated by the significant compute and engineering burden of external agentic complexity. Worse, this heavy investment does not yield proportional gains, often witnessing a "seesaw effect" on perception and reasoning. This motivates a fundamental rethinking of the true bottleneck. In this paper, we argue that the root cause of this trade-off is an ambiguity in modality credit assignment: when a VLM fails, is it due to flawed perception ("bad seeing") or flawed logic ("bad thinking")? To resolve this, we introduce a reinforcement learning framework that improves perception-reasoning synergy by reliably rewarding the perception fidelity. We explicitly decompose the generation process into interleaved perception and reasoning steps. This decoupling enables targeted supervision on perception. Crucially, we introduce Perception Verification (PV), leveraging a "blindfolded reasoning" proxy to reward perceptual fidelity independently of reasoning outcomes. Furthermore, to scale training across free-form VL tasks, we propose Structured Verbal Verification, which replaces high-variance LLM judging with structured algorithmic execution. These techniques are integrated into a Modality-Aware Credit Assignment (MoCA) mechanism, which routes rewards to the specific source of error – either bad seeing or bad thinking

– enabling a single VLM to achieve simultaneous performance gains across a wide task spectrum.

## 1. Introduction

Humans navigate the world through a seamless, synergistic interplay between visual perception (seeing) and cognitive reasoning (thinking). We do not "see" once and then "think" in isolation; rather, we continuously re-interrogate visual evidence to form, verify, and refine hypotheses. This tight loop, where cognition guides perception and perception grounds cognition – a dynamic we term **perception-reasoning synergy** – is the hallmark of robust, generalizable intelligence.

Achieving this synergy has been the central pursuit of advanced Vision-Language Models (VLMs), fueling distinct architectural paradigms. Initial architectures (e.g., Qwen-VL) focused on implicitly fusing visual tokens and textual embeddings, relying on the model's static textual reasoning to process this fused information (Liu et al., 2023; Xu et al., 2025a; Wang et al., 2024; Bai et al., 2025). More recent works, aspiring toward "thinking with images," have embraced active perception by incorporating visual operations (Wang et al., 2025d), more closely reflecting the human synergistic loop.

These active-perception models, however, are often implemented through complex, multi-turn function-calling or agentic frameworks. This approach introduces both significant compute and engineering burden, such as implementing multi-turn RL training and handling asynchrony for long-tailed episodes, and often yields a "seesaw effect," where marginal visual gains come at the expense of reasoning capabilities (Jiang et al., 2025; Wang et al., 2025d). We posit that a crucial yet under-explored bottleneck, unique to vision-language reasoning, is the lack of direct supervision signals over perception. Current training paradigms fail to provide modality-aware feedback for vision-language reasoning. Was the reasoning error caused by flawed perception (bad seeing) or flawed logic (bad thinking)?

This ambiguity in modality credit assignment is overlooked and challenging because a VLM's perception is inherently difficult to access: it is either (a) buried in latent-state activations or (b) inextricably entangled with the reasoning

[1]Hong Kong University of Science and Technology (HKUST) [2]Tsinghua University [3]Zhejiang University [4]Meituan, China [5]University of Waterloo . Correspondence to: Haozhe Wang <jasper.whz@outlook.com>.

*Proceedings of the 43rd International Conference on Machine Learning*, Seoul, South Korea. PMLR 306, 2026. Copyright 2026 by the author(s).

components in the final textual output. This entanglement makes it nearly impossible to supervise perception independently of logic. This prompts us a question:

*Can we improve multimodal reasoning by externalizing perception and directly rewarding it?*

Our answer is yes. In this work, we explore a simple way to externalize perception, leveraging instruction-following capabilities to enforce a structural separation in the generation process: perception-centric blocks (e.g., <recognition>) that isolate visual evidence are interleaved with reasoning-centric blocks (e.g., <think>) (Fig. 2). This choice transforms the opaque "black box" of VLM perception into a transparent sequence, exposing the specific locus of error – bad seeing or bad thinking – and allowing us to reframe the intractable problem of joint supervision into two solvable, component-level challenges:

1. **The Perception Verification:** How do we specifically supervise the quality of the model's perception, independent of the final reasoning step?
2. **The Outcome Verification:** To achieve improvement across a wide task spectrum, How do we reliably reward free-form answers, avoiding the brittleness of regex and the high variance of standard LLM judges?

Given isolated perception explicitly grounded in the textual space, we tackle the first challenge: the lack of ground truth for intermediate visual descriptions. We introduce **Perception Verification via Proxy**. Our key insight is that in explicit vision-language reasoning, visual details serve as discrete premises for logical deduction; therefore, perception sufficiency can be measured by reasoning feasibility. We implement this as a "blindfolded reasoning" test: we feed the VLM's grounded visual evidence to a strong, text-only reasoner while withholding the image. If this text-only proxy can correctly solve the user's question using *only* the VLM's descriptions, it demonstrates that the perception block has successfully captured the sufficient statistics required for the task. This signal rewards the VLM not for generic captioning, but for extracting the precise visual facts necessary to support downstream reasoning.

While the proxy validates perception, we must also ensure robust supervision for the final reasoning outcomes. Existing evaluators suffer a dichotomy: flexible LLM judges exhibit high variance, while rigid rule-based verifiers are brittle. We bridge this gap with **Structured Verbal Verification (SVV)**. Rather than asking an LLM for a holistic and subjective "judgment," we provide a structured natural language algorithm—a verification protocol—and instruct the judge to explicitly "execute" this protocol step-by-step. This shifts the verifier's role from subjective estimation to structured execution, significantly reducing variance and ensuring that our reward signal remains reliable even for

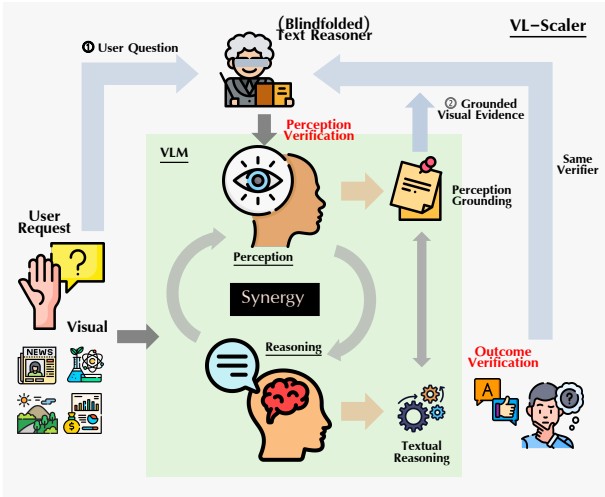

*Figure 1.* **Overview of MoCA.** MoCA identifies "bad seeing" from "bad thinking" by separating Perception and Reasoning. We introduce Perception Verification (PV, top loop), which uses a "blindfolded" text reasoner proxy to reward the VLM's perception independently of its reasoning. This, combined with Outcome Verification (OV, bottom loop), rewards for better perception-reasoning synergy. To render an economical implementation, we use the same verifier for PV and OV.

complex, free-form responses.

we unify these techniques into a **Modality-aware Credit Assignment (MoCA)** mechanism. By routing these granular rewards to their specific source—punishing "bad seeing" or "bad thinking" distinctly—we dismantle the traditional "seesaw effect" where gains in one modality come at the expense of the other. Conceptually, our approach mirrors the agentic "thinking with images" paradigm but internalizes it. Rather than invoking external tools via slow, multi-turn function loops, our model treats its own perception capabilities as an internal function, calling upon it to retrieve visual evidence before reasoning. This offers a lightweight, scalable path to perception-reasoning synergy that is orthogonal to, yet distinct from, existing external agentic frameworks.

We conduct extensive empirical validation across a comprehensive suite of multimodal benchmarks. Our results demonstrate that the proposed approach outperforms state-of-the-art vision-language reasoners across a wide task spectrum spanning from perception-intensive reasoning to rich-modality reasoning.

**Contributions.** We fundamentally rethinks VLM training by shifting from holistic outcome supervision to decoupled Modality-aware Credit Assignment (MoCA).

- We enforce an explicit architectural decoupling of perception and reasoning, transforming the VLM from an opaque box into an interpretable, modular reasoner.
- We introduce **Perception Verification via Proxy**, a

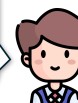

**<image>** What is the cost limit for the renewal of the property management agreement with MB Real Estate Services LLC for the one-year term?

**reasoning**

**<think>**First, the document appears to be a formal report ... The document outlines .. Let me break down the user's question and think about where I might find this information in the document.**</think>**

**reasoning**

**<think>**The user wants to know .. I'll need to scan the document for ... I expect to find this information in ...**</think>**

**perception**

**<recognition>**Scanning through the document, I locate the relevant information in the section that states: \"Approve the renewal of the agreement with MB ...**</recognition>**

**reasoning**

**<think>**Great! The document explicitly mentions the cost limit for the renewal of the agreement. According to the text ... However, I should ensure ... elsewhere in the document.**<think>**

**perception**

**<recognition>**I've reviewed the rest of the document, and there are no other mentions of different cost limits for the renewal. The specified amount remains consistent throughout the document....**</recognition>**

**<answer>**The cost limit for the renewal of the property management agreement with MB Real Estate Services LLC for the one-year term is \\\boxed{$374,355.00}.**</answer>**

*Figure 2.* Illustration of Interleaved Perception-Reasoning.

novel "blindfolded reasoning" paradigm that validates visual fidelity by checking if a text-only reasoner can solve the task using the model's descriptions alone.
- We propose **Structured Verbal Verification (SVV)** for robust free-form evaluation, enabling us to internalize the benefits of agentic "thinking" without external computational overhead.

## 2. Methodology

Our approach targets at scaling VL reasoning across a wide task spectrum. To achieve this, we focus on addressing the perception supervision and outcome verification problem, optimizing the *internal* perception-reasoning synergy. Below we first formalize the problem to expose its core challenge and then introduce our solution.

### 2.1. Problem Formulation

We formulate the task of generating an answer $Y$ given an image $V$ and question $Q$ as a sequential decision-making process. The objective is to learn a single, generalizable

policy $\pi_\theta$ (the VLM) that generates a reasoning trajectory $\tau = (a_1, a_2, ..., a_T)$ to arrive at the correct final answer.

This process is a (Partially Observable) Markov Decision Process (POMDP) (Sondik, 1971), where the state $s_t = (V, Q, a_1, ..., a_{t-1})$ represents the history of actions. Our central insight is to deliberately decompose the VLM's native auto-regressive action space $\mathcal{A}$ into two distinct, component-level sets of actions:

- **Perception Actions ($\mathcal{A}_P$):** are Perception-centric text blocks (e.g., <recognition>...</recognition>) that serve to ground visual facts and extract relevant evidence from the image $V$. We focus on multimodal task that requires explicit reasoning and posit that grounded perception text act as the discrete premises required for the subsequent logical deduction.
- **Reasoning Actions ($\mathcal{A}_R$)** are reasoning-centric text blocks that understand user requests, decompose problems, form hypotheses, draw inferences (e.g., <think>...</think>).

The policy $\pi_\theta$ is thus an auto-regressive model $p(a_t|s_t)$ that generates an interleaved sequence of these perception and reasoning actions.

This explicit decomposition exposes the fundamental bottleneck to perception-reasoning synergy: **Ambiguous Credit Assignment**. This challenge is distinct from the standard process supervision challenge of validating reasoning in text-only LLMs. In purely textual tasks, intermediate steps can be verified against textual logic. Our **modality credit assignment** problem is unique: the perception actions ($a_p$) are grounded in the *visual inputs* ($V$), not text. This makes it almost impossible to supervise perception quality by simply looking at the final answer, as a failure could stem from flawed perception ('bad seeing') or flawed logic ('bad thinking').

### 2.2. Eliciting Perception-Reasoning Decomposition

To directly supervise the perception of VLMs, we externalize the perception-reasoning synergy inherent in the model, in the interleaved perception-reasoning format (see Figure 2). To achieve this, we leverage the instruction-following capabilities of modern instruction-tuned VLMs.

We provide the model with a single, task-agnostic instruction as part of its system prompt (Figure 3). This instruction compels the model to "think step-by-step" by externalizing its process into perception-centric <recognition>...</recognition> blocks and reasoning-centric <think>...</think> blocks. This instruction-driven approach avoids the need for a large-scale, synthesized trajectory dataset and allows the subsequent reinforcement learning stage to optimize the model's native decomposition capabilities from its pre-trained state.

```
You will be given images and a question. You are required to
provide a step-by-step thinking process (inner monologue) by
interleaving <think> and <recognition> tags, culminating in
a final <answer>.

Throughout your response, you will interleave two tag block:
• `<think>` and `</think>` block: Use this tag for planning,
  reasoning, self-reflection, and synthesizing information.
  This is where you analyze the query, formulate a plan,
  connect evidence, and draw conclusions.
• `<recognition>` and `</recognition>` block: Use this tag
  *exclusively* for information extraction and grounding.
  You must meticulously list or quote the *exact* relevant
  text snippets, data points, and details from the images
  that you are using to support your reasoning.

Your response must follow this interleaved cycle of
`<think>` and `<recognition>`. Finally, you need to indicate
the final answer with the required `\\boxed` command. A
`<answer>` and `</answer>` block is highly recommended to
clearly decorate the final answer.

Question: \{question\}
```

*Figure 3.* Instruction to elicit interleaved perception-reasoning.

### 2.3. Rewarding Perception: The "Blindfolded Reasoner" Test

With perception now an explicit output (`<recognition>`...), our goal is to evaluate its quality. The core challenge is that *there is no ground truth for an intermediate "perceptual thought"*. We cannot use a static label. Therefore, we must measure its quality functionally. Our central insight is that to assign credit, we can *isolate* the perception component's contribution from the reasoning process.

We implement this isolation using a strong, text-only reasoner (e.g., Qwen2.5-Instruct-14B) as the proxy for a reasoning-perfect oracle. Our premise is that the reasoning component of most vision-language tasks, once visual facts are established, is less complex than pure, abstract reasoning. We call this the "Blindfolded Reasoner" Test:

1. The VLM generates a perception action $a_p$ (e.g., "After scanning the document, I locate the relevant figure which shows that ...").
2. We provide this strong reasoner with the original question $Q$ and all the VLM's generated perception text $\{a_p\}$, **withholding the image** $V$.
3. We obtain a final answer from the reasoner using only these grounded visual evidences.

If this "image-blind" reasoner succeeds, the collection $\{a_p\}$ is demonstrated to be a *sufficient statistic* for the image $V$ relative to the question $Q$. Conversely, if the reasoner fails, we assign the blame to the flawed $\{a_p\}$ perception grounding. This yields the targeted, modality-level reward signal, $R_P(\tau)$, that we need.

We call this framework **Perception Verification (PV)**. This technique is not ad-hoc; it serves as a functional proxy for

the Information Bottleneck (IB) principle (Tishby et al., 2000; Alemi et al., 2016). It rewards the perception block collection $\{a_p\}$ that is maximally informative about the answer $Y$ (by being sufficient for the oracle) while being a minimal (textual) representation of the image $V$, i.e., $\min_{p(A_p|V)} \quad I(V; A_p) - \beta I(A_p; Y)$. To ensure minimalism, we explicitly penalize perception blocks $\{a_p\}$ that exceed a predefined token limit (e.g., 800 tokens).

**Validity of Textual Grounding.** One might question the feasibility of perception grounding, given that certain visual patterns (e.g., mazes) cannot be trivially converted to text. However, within the scope of explicit multimodal reasoning (System 2), a solver must extract discrete task-relevant variables (e.g., "the slope is positive") to form a logical chain. As a result, perception grounding, the identification and grounding of any visual feature critical to the solution, serves as a premise for symbolic reasoning. To this end, the Blindfolded Reasoner functions as a sufficiency test: it rewards the VLM not for describing image pixels, but for extracting the sufficient statistics required to satisfy the specific reasoning query.

**Reward Definition.** Based on the proxy's performance, we assign a binary perception reward $R_P(\tau) \in \{0, 1\}$ to the trajectory:

$$R_P(\tau) = \begin{cases} 1, & \text{if } \text{Proxy}(Q, \{a_p\}) = Y_{\text{correct}} \\ 0, & \text{otherwise} \end{cases}$$

This discrete signal is then used to gate the credit assignment in our MOCA framework.

**Discussion: Robustness to Oracle Fallibility.** We acknowledge that the text-only reasoner is often a high-capability proxy and not a perfect, infallible reasoner. It can, therefore, introduce false-negative noise (i.e., failing to reason from a truly sufficient $a_p$), leading to an incorrect penalty. While our MoCA objective is designed to mitigate the impact of this noise during failed trajectories, we posit that such instances are infrequent. We include a human-evaluation study in the experiment section to quantify the agreement rate between our PV oracle and human judgment on "perceptual sufficiency," thereby validating the reliability of $R_P$ as a reward signal.

### 2.4. Structured Verbal Verification

Our "blindfolded reasoner" (Section 2.3) provides the perception reward, $R_P$. To solve the full credit assignment problem, we also need a reliable reward for the final outcome, $R_O(\tau)$. This is a non-trivial challenge for a generalist model handling free-form answers. Existing methods typically fall into two camps: Rigid Rule-Based Verifiers (e.g., regex, keyword matching), which are precise but brittle, and LLM Prompting (e.g., "Is this equivalent?"), which is flexible but suffers from high variance.

```
Evaluate the semantic similarity between a Model Response and a Ground Truth Answer.
Question: {question}
Ground Truth Answer: {ground_truth}
Model Response: {prediction}

## EVALUATION ALGORITHM
### STEP 1: Identify Answer Type
...

### STEP 2: Extract Boxed Content
...

### STEP 3: Apply Type-Specific Scoring
...

#### IF Type D (Sentence/Paragraph):
**3.1** Reconstruct reference answer:
Write: "Reference answer sentence: [complete sentence based on question and GT]"

**3.2** Extract implied answer from model:
Write: "Model's implied answer: [sentence/meaning extracted from model response]"

**3.3** Semantic comparison:
Analyze whether both sentences convey the same meaning. Consider:
...
Write: "Analysis: [detailed comparison]"

**3.4** Assign semantic similarity score:

Write: "Type C Score: [score]"

### STEP 4: Final Judgment
...

Now execute the algorithm above step by step.
```

*Figure 4.* Instruction Prompt of Structured Verbal Verification.

To bridge this gap, we propose Structured Verbal Verification (SVV). Rather than asking an LLM Judge to vaguely "judge" the final answer $Y$ or asking it to write Python code, we provide a structured natural language algorithm—a verification protocol—and instruct the judge to explicitly "execute" this protocol step-by-step (as shown in Figure 4). This protocol forces the verifier to decompose the evaluation into distinct stages: (1) identifying the answer type, (2) extracting content, (3) reconstructing the reference, and (4) performing type-specific semantic comparison. This shifts the verifier's role from subjective estimation to structured execution, significantly reducing variance and ensuring that our reward signal remains reliable even for complex, free-form responses.

A critical aspect of SVV is its scalability; we do not handcraft unique protocols for every question. Instead, we employ a universal verification protocol that covers the majority of verification cases (Figure 4). By forcing the LLM Judge to "execute" this universal program verbally, we obtain the robust, low-variance outcome reward $R_O$ required for stable RL training.

### 2.5. MoCA: Resolving the "Bad Seeing vs. Bad Thinking" Ambiguity

Given two core reward signals – $R_O$ for outcome correctness and $R_P$ for perceptual fidelity, we now integrate them into a coherent reinforcement learning objective. To achieve perception-reasoning synergy, we define trajectory-level return as the sum of the outcome and perception rewards:

$$R(\tau) = R_O(\tau) + \lambda R_P(\tau)$$

Following standard practice in Group Relative Policy Op-timization (GRPO) (Shao et al., 2024), we estimate advantages by computing a group-wise reward baseline. For a given query, we sample a group of $k$ trajectories $\{\tau_1, ..., \tau_k\}$ from our policy $\pi_\theta$. The advantage $A_{\tau,t}$ for any token $t$ in a trajectory $\tau$ is its outcome reward $R_O(\tau)$ normalized against the group's mean reward:

$$A_{\tau,t} = R(\tau) - \frac{1}{k} \sum_{j=1}^{k} R(\tau_j)$$

The bottleneck lies in the failure cases where the outcome is incorrect ($R_O = 0$). Standard GRPO applies the calculated advantage $A_{\tau,t}$ (which is negative) uniformly across the sequence, punishing all tokens regardless of whether the error stemmed from perception or reasoning. This creates a "see-saw effect": the model risks "unlearning" correct perception behaviors simply because the subsequent reasoning failed.

To resolve this, our Modality-Aware Credit Assignment (MOCA) mechanism uses the binary perception reward $R_P \in \{0,1\}$ as a gate to route credit assignment. We define the modified advantage $A_{\tau,t}^{\text{MOCA}}$ specifically for failed trajectories ($R_O = 0$) to distinguish between two scenarios:

- **Case 1, "Bad Thinking".** If the outcome is wrong ($R_O = 0$) but perception was verified as correct ($R_P = 1$), the error stems from reasoning. We "protect" the innocent perception tokens ($\tau_P$) by dampening the penalty with a positive protection term:

$$A_{\tau,t} + \alpha_{\text{protect}} \cdot |A_{\tau,t}|.$$

  This prevents the gradient update from degrading valid visual grounding capabilities.

- **Case 2, "Bad Seeing".** If the outcome is wrong ($R_O = 0$) and perception was also failed ($R_P = 0$), the error likely stems from the visual grounding. We amplify the penalty for these perception tokens:

$$A_{\tau,t} - \alpha_{\text{punish}} \cdot |A_{\tau,t}|.$$

This mechanism precisely routes blame and protection to the correct components. Furthermore, the "protect" mechanism (Case 1) is non-trivial: it makes our framework robust to the potential false-negative noise from our perception proxy (discussed in Section 2.3), preventing the model from unlearning good perception due to imperfect verification. By scaling up the data and verifier capabilities, this approach offers a generalizable solution to the credit assignment problem in vision-language reasoning.

## 3. Experiments

To empirically validate our framework, we design experiments to answer three key research questions (RQs):

- **(RQ1) Main Claim:** Does MoCA achieve simultaneous performance gains across perception- and

*Table 1.* Main Results.

| Model | Size | Perception-Centric | | | Rich Modalities | | | Reasoning-Centric | | |
|---|---|---|---|---|---|---|---|---|---|---|
| | | V* | HRBench | InfoVQA | DUDE | SlideVQA | MMLong | MMMU | EMMA | MathVista |
| *General-ability model* | | | | | | | | | | |
| GPT-4o | - | 45.0 | 65.0 | 80.7 | 52.7 | 53.1 | 42.3 | 51.9 | 32.7 | 63.4 |
| GPT-4o-mini | - | 50.8 | 48.0 | 83.3 | 46.5 | 48.1 | 28.6 | 45.1 | 27.3 | 60.0 |
| Claude-3.5 | - | - | - | 74.3 | - | - | - | 68.3 | 38.1 | 67.7 |
| Qwen2.5-VL-Instruct | 72B | 81.2 | 73.4 | 84.3 | 44.5 | 38.8 | 24.9 | 67 | 38.5 | 74.8 |
| Llava-OV | 7B | 72.8 | 64.7 | 88.3 | 38.1 | 35.0 | 19.5 | 48.8 | 18.3 | 63.2 |
| Qwen2.5-VL-Instruct | 7B | 71.4 | 69.2 | 80.7 | 41.8 | 38.9 | 21.2 | 54.3 | 21.5 | 68.2 |
| *Capability-Enhanced model* | | | | | | | | | | |
| VL-Rethinker | 7B | 68.2 | 65.0 | 79.5 | 39.1 | 38.5 | 20.9 | 56.7 | 29.7 | 74.9 |
| R1-VL | 7B | 60.3 | 53.1 | 78.0 | 25.4 | 22.1 | 12.7 | 7.8 | 8.3 | 63.5 |
| Pixel Reasoner | 7B | 84.3 | 72.8 | 86.4 | 44.5 | 42.1 | 22.0 | 50.8 | 19.8 | 65.3 |
| DeepEyes | 7B | 88.9 | 73.1 | 87.7 | 35.2 | 33.0 | 17.5 | 45.2 | 18.1 | 64.9 |
| Docopilot | 8B | 40.1 | 48.3 | 75.0 | 40.7 | 35.7 | 28.8 | 36.6 | 12.1 | 45.0 |
| mPLUG-Owl3 | 7B | 64.5 | 60.1 | 76.3 | 39.5 | 48.8 | 21.0 | 42.9 | 15.0 | 50.7 |
| *Our Models* | | | | | | | | | | |
| **MoCA** | 7B | 86.6 | 74.2 | 87.0 | 45.1 | 58.3 | 33.1 | 54.8 | 31.3 | 73.8 |

reasoning-intensive tasks? How does it compare against existing state-of-the-art models?

- **(RQ2) Component Value:** Are the core components of our approach, Perception Verification (PV), Structured Verbal Verification (SVV), and modality-aware credit assignment – all necessary for this success?

- **(RQ3) Reward Reliability:** Is our $R_P$ signal from PV a reliable proxy for "perceptual reliability," especially given potential oracle fallibility?

**Training Datasets.** Our training corpus is curated to cover a diverse task spectrum, essential for training a generalizable model. It includes: (1) **Instruction & Reasoning Data:** A combination of STEM-focused reasoning tasks from ViRL39K (Wang et al., 2025b) and general-purpose visual instructions from VisualWebInstruct-Verified (Jia et al., 2025). (2) **Perception-Intensive Data:** A collection of visually-rich, fine-grained queries sourced from the Pixel Reasoner dataset (Wang et al., 2025d). (3) **Modality-Rich Data:** A novel, curated collection of vision-language queries that involves reasoning over rich modalities, such as the interplay between text, figures, diagrams, layouts, and tables. This dataset is gathered by crawling and processing documents from arXiv, newspapers, and infographics. We provide a further breakdown in the appendix.

**Evaluation Benchmarks and Protocols.** To comprehensively measure performance, we categorize benchmarks into three distinct groups.

- **Perception-Intensive Benchmarks:** We use V* (Wu & Xie, 2023), HRBench (Wang et al., 2025f), and InfoVQA (Mathew et al., 2021) to evaluate fine-grained

visual perception and grounding.

- **Reasoning-Intensive Benchmarks:** We use Math-Vista (Lu et al., 2024), MMMU (Yue et al., 2024), and EMMA (Hao et al., 2025) to test complex, multi-step reasoning grounded in visual information.

- **Rich-Modality Benchmarks:** We use DUDE (Landeghem et al., 2023), SlideVQA (Tanaka et al., 2023), and MMLongBench-Doc (Ma et al., 2024) to assess performance on document-related tasks requiring reasoning over rich, structured modalities, including text, figures, diagrams, layouts.

**Baselines.** We compare MoCA against a comprehensive suite of models, including capability-enhanced open-source models and top-tier commercial systems. To save space, we refer interested readers to the appendix for performance comparison with more models.

- **General-ability Models:** GPT-4o, 4o-mini (OpenAI et al., 2024), Llava-OV (Li et al., 2024), Qwen2.5-VL-Instruct-7B (our base model) (Bai et al., 2025).

- **Capability-Enhanced Models:** VL-Rethinker (Wang et al., 2025b) and R1-VL (Zhang et al., 2025a), Pixel Reasoner (Wang et al., 2025d), Docopilot (Duan et al., 2025), mPLUG-Owl3 (Hu et al., 2024).

**Implementation Details.** Unless otherwise specified, all open-source methods are built upon the Qwen2.5-VL (7B-Instruct) base model to ensure a fair comparison. For our novel reward mechanisms, we use Qwen2.5-Instruct-14B as the model for both the text-only reasoner (PV) and the Verification by Structured Verbal Verification.

## 3.1. Main Results (RQ1)

We present our main quantitative results in Table 1, providing strong empirical evidence for our central claim (RQ1): MoCA successfully resolves the perception-reasoning trade-off by achieving simultaneous, significant performance gains across all task categories, countering the "seesaw effect" that plagues other methods. Compared to its Qwen2.5-VL-Instruct (7B) base, MoCA demonstrates massive, broad-spectrum improvements. This clearly validates that our framework, by explicitly supervising perception and reasoning, successfully optimizes for their synergy without degrading one capability to enhance another.

Furthermore, MoCA establishes itself as a top-performing 7B model against specialized and proprietary systems. It proves highly competitive with perception-centric models like DeepEyes while setting a new standard in rich-modality tasks (58.3 on DUDE). Most notably, MoCA-7B can even surpass proprietary models, e.g., perception-and reasoning-intensive tasks over GPT-4o. This balanced and elevated performance profile validates MoCA as a robust model that scales across the full spectrum of vision-language tasks.

## 3.2. Ablation Studies: Isolating Component Contributions (RQ2)

To validate each components of our framework, we conduct a comprehensive ablation study, with results presented in Table 2. We compare the full MoCA against variants to isolate the contributions of Perception Verification (PV), Structured Verbal Verification, and our Modality-aware Credit Assignment (MoCA) logic.

**Total Benefit of RL Optimization.** First, we compare **MoCA (Full)** against the **Instruction-Only (No RL)** baseline. This baseline uses our decompositional prompt but undergoes no RL optimization. The results show a massive performance gap across all categories. Merely prompting the model for decomposition brings a performance drop compared to the base model, because the model is not familiar with the decomposed perception-reasoning format. This result also confirms that the proposed overall framework leads to significant improvement across the board.

**Benefits of Perception Verification (PV).** The **w/o PV** ($R_O$ **only**) variant removes the perception reward $R_P$ and relies only on the final outcome reward $R_O$. This equals the standard GRPO approach with the VP outcome reward. The results show performance drops significantly on perception-intensive tasks, e.g., -6.9 points on V*, -4.7 on HRBench, and -4.1 on InfoVQA, indicating the effectiveness of properly rewarding the perception during vision-language reasoning. This confirms our core hypothesis: lacking targeted perception supervision is a crucial bottle-

neck in vision-language reasoning. Without $R_P$, the model cannot distinguish "bad seeing" from "bad thinking" and fails to optimize its perceptual capabilities. In the meantime, reasoning-centric tasks are largely unaffected, because these tasks usually do not require complex perception capabilities.

**Benefits of Modality-aware Credit Assignment (MoCA).** The baseline **w/o MoCA** ($R_O + \lambda R_P$) uses both reward signals but naively combines them, which we argued creates ambiguity in failure cases. The results strongly supports our claim. This simple reward shaping variant performs worse than our full model, lagging by nearly 3 points on perception-intensive tasks and 2 points on rich-modality tasks. This demonstrates that our MoCA logic is non-trivial and essential. By protecting "good" perception tokens and punishing "bad" ones during failed trajectories, MoCA correctly assigns blame and prevents the model from "unlearning" good perception habits, which the simple shaping approach fails to do.

**Benefits of Structured Verbal Verification.** Finally, the **w/o VP+PV (LLM Judge)** variant uses the outcome reward from LLM judge and applies the standard GRPO. The results show a consistent performance drop from **w/o PV** baseline. The hurt of high-variance reward is indeed surprising. We find that the model tends to reward hacking by exploiting the LLM judge, making the RL training less stable and misleading the model into irrelevant reward hacking behaviors.

## 3.3. Reward Reliability (R3)

**Perception Verification.** To validate the reliability of our Perception Verification (PV) proxy, we test its reliability against human judgment. We randomly sampled $N = 979$ data points, each containing a (question, VLM-generated perception text, golden answer) triplet. Three human annotators were asked to judge if the perception text alone was "Sufficient" or "Insufficient" to logically deduce the golden answer. We refer interested readers to the supplementary materials for a full breakdown of the experimental setup, human annotation process, and detailed results.

We then compared the PV oracle's verdict (using Qwen2.5-Instruct-14B) against the human majority vote. As shown in Table 3, the oracle achieved reasonable fidelity with human judgment (86.31% accuracy, Cohen's $\kappa = 0.707$, indicating "Substantial" agreement). Crucially, the analysis of disagreements reveals the oracle's primary failure mode is conservative False Negatives (9.19%)—where it fails to reason from text humans found sufficient—over False Positives (4.49%). This result validates our oracle as a reliable proxy and confirms the necessity of our Decomposed Credit Assignment (DCA) objective (Section 2.5), which is specifically designed to handle and mitigate this exact type of

*Table 2.* Ablation Studies. We analyze the contribution of each component of MoCA (Full). All variants are trained with the same data and base model. $\Delta$ indicates the performance drop from the full model.

| Model | Perception-Centric | | | Rich Modalities | | | Reasoning-Centric | | |
|---|---|---|---|---|---|---|---|---|---|
| | V* | InfoVQA | HRBench | MMLong | SlideVQA | DUDE | EMMA | MMMU | MathVista |
| Full Model | 86.6 | 87.0 | 74.2 | 33.1 | 58.3 | 45.1 | 31.3 | 54.8 | 73.8 |
| Instruction-Only (No RL) | 68.3 | 78.1 | 66.5 | 17.1 | 39.3 | 37.7 | 20.1 | 49.9 | 65.7 |
| w/o PV ($R_O$ only) | 79.7 | 82.3 | 70.1 | 30.3 | 54.1 | 42.5 | 30.9 | 55.3 | 74.4 |
| w/o MoCA ($R_O + \lambda R_P$) | 83.1 | 84.3 | 72.5 | 31.1 | 56.3 | 43.7 | 30.5 | 54.6 | 74.1 |
| w/o SVV+PV (LLM Judge) | 78.4 | 80.9 | 69.7 | 27.3 | 54.5 | 38.9 | 29.1 | 52.3 | 72.1 |

"fallible oracle" noise.

*Table 3.* PV Oracle vs. Human Majority Confusion Matrix ($N = 979$).

| | | Human Majority Verdict | |
|---|---|---|---|
| | (% of Total) | Sufficient | Insufficient |
| **PV Oracle** | **Sufficient** | 550 (56.18%) | 44 (4.49%) |
| **Verdict** | **Insufficient** | 90 (9.19%) | 295 (30.13%) |
| **Total** | | 640 (65.37%) | 339 (34.63%) |

**Outcome Verification.** We validate our Structured Verbal Verification method against two common baselines: a **Rigid Rule** verifier (regex, exact match) and a standard **LLM Prompting** verifier (e.g., "Are these answers semantically equivalent?"). We created a "VP-Challenge-Set" ($N = 273$) of (model answer, gold answer) pairs, specifically including difficult semantic rephrasings and answers with subtle errors. We measure both accuracy against human-annotated labels and **Consistency** (the percentage of identical verdicts over 5 runs at $T = 0.7$). Further details on the challenge set construction and the universal verbal program are available in the supplementary materials.

As shown in Table 4, the baselines fail in predictable ways. *Rigid Rule* has high precision but fails on all semantic rephrasings, resulting in poor recall. *LLM Prompting* is flexible but high-variance and unreliable, achieving only 78.6% consistency. Our VP method achieves the highest accuracy (91.9%) and F1-score (92.7%) by transforming the vague "judgment" into a structured "execution." This makes it a robust, low-variance reward signal, achieving 92.3% consistency, which is critical for stable RL training.

*Table 4.* Verifier performance on the VP-Challenge-Set ($N = 273$). VP provides the best balance of accuracy and reliability.

| Verifier | Acc. (%) | F1 (%) | Consistency (%) |
|---|---|---|---|
| Rigid Rule | 67.0 | 58.7 | 100.0 |
| LLM Prompting | 79.1 | 82.4 | 78.6 |
| **VP (Ours)** | 91.9 | 92.7 | 92.3 |

## 4. Related Work

**Vision-Language Reasoning.** Recent advancements in Large Vision-Language Models (LVLMs) have significantly improved reasoning capabilities by aligning visual encoders with LLMs (Liu et al., 2023; Wang et al., 2024; Bai et al., 2025; Dai et al., 2023; Wei et al., 2022). While Chain-of-Thought (CoT) prompting has unlocked complex reasoning in text, its application in vision remains challenging due to perceptual hallucinations (Wei et al., 2023; Zhu et al., 2025). Recent works attempt to mitigate this via supervised fine-tuning and reinforcement learning or distillation (Jia et al., 2025; Wei et al., 2022; Wang et al., 2025b; Ma et al., 2025; Ding et al., 2025), or align modalities via perplexity, logit differences (Zhou et al., 2024; Leng et al., 2024; Zhu et al., 2024). However, these methods often treat perception and reasoning as an entangled latent process, making it difficult to diagnose whether errors stem from perception or reasoning.

**Synergy via Agentic Function-Calling.** To bridge the perception-reasoning gap, several approaches adopt agentic frameworks that utilize external tools or multi-turn verification loops (Wang et al., 2025d; Zhang et al., 2025b; Ma et al., 2026). While effective, these systems incur significant latency and engineering complexity. VL-Scaler proposes an orthogonal direction: internalizing this synergy. By explicitly interleaving perception and reasoning tokens within a single autoregressive generation, we retain the efficiency of end-to-end models while capturing the interpretability and robustness of agentic workflows.

**Reward Engineering and Credit Assignment.** Reinforcement learning (RL) has become standard for aligning VLMs (Bai et al., 2025; Team et al., 2025; Wang et al., 2025d). However, standard outcome-based supervision (e.g., RLHF, DPO (Ouyang et al., 2022; Rafailov et al., 2023)) suffers from ambiguous credit assignment in multimodal contexts (Wang et al., 2025e). Unlike methods that rely on high-variance LLM judges or rigid regex (Shao et al., 2024), our Structured Verbal Verification and Perception Verification introduce deterministic, modality-aware reward signals that explicitly decouple and supervise the visual and logical reasoning components.

## Impact Statement

This paper presents work whose goal is to advance the field of Machine Learning. There are many potential societal consequences of our work, none which we feel must be specifically highlighted here.

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

## A. Training Setup and Implementation Details

We utilize `Qwen2.5-VL-Instruct-7B` as the base policy model $\pi_\theta$, while employing the text-only `Qwen2.5-Instruct-14B` for both perception verification and outcome verification. Training is conducted using Group Relative Policy Optimization (GRPO) on a cluster of $16 \times$ NVIDIA H100 (80GB) GPUs, taking approximately 16 hours for convergence. Key hyperparameters include a learning rate of $1e^{-6}$ with a cosine decay scheduler, a global batch size of 256 accumulated gradient steps and 512 QA rollouts per policy sync, a group size of $G = 8$, and a KL divergence coefficient of $\beta = 0.0$. The reward signal combines an outcome weight of $1.0$ and a perception weight of $\lambda = 0.3$, utilizing MoCA penalties with $\alpha_{protect} = 0.2$ and $\alpha_{punish} = 0.2$. The hyperparameters are selected based on an initial hyperparameter sweep using smaller 3B models.

## B. Benchmark Categorization and Details

We categorize benchmarks based on the primary bottleneck for model performance.

**1. Perception-Intensive Benchmarks.** These tasks require fine-grained visual recognition, OCR, or grounding, where the reasoning is straightforward once the visual element is identified.

- **V\* (V-Star):** Focuses on guided visual search and spotting small details in high-resolution images.
- **HRBench (4K):** Evaluates hyper-resolution perception, critical for detecting minute objects often lost in standard resizing.
- **InfoVQA:** Requires reading and extracting specific text from large high-resolution document images and infographics.

**2. Reasoning-Intensive Benchmarks.** These tasks provide visually distinct information but require complex logical chains, mathematical calculation, or domain knowledge.

- **MathVista:** Requires multi-step mathematical reasoning and geometric logic.
- **MMMU:** A massive multi-discipline benchmark requiring expert-level knowledge and logic across college-level subjects.
- **EMMA:** Specifically designed to test "Chain-of-Thought" reasoning in multimodality.

**3. Rich-Modality Benchmarks.** These tasks occupy the intersection, requiring the model to parse complex structures (layouts, tables, diagrams) and reason over them.

- **DUDE:** Document Understanding Dataset and Evaluation, involving multi-page documents with diverse layouts.
- **SlideVQA:** Requires retrieving information from a deck of presentation slides, testing layout understanding and sequence logic.
- **MMLongBench-Doc:** Tests the model's ability to handle long-context multimodal documents.

## C. Modality-Rich Training Dataset

Table 6 details the composition of our curated training dataset, including STEM-related reasoning-intensive QAs from VL-Rethinker (Wang et al., 2025b) and VisualWebInstruct (Jia et al., 2025), high-resolution natural scene QAs from Pixel Reasoner (Wang et al., 2025d), and the collected modality-rich queries. We show a few examples of the curated modality-rich queries in Fig. 5. We will release the code and dataset upon paper acceptance.

*Table 5.* Extended Main Results.

| Model | Size | Perception-Centric | | | Rich Modalities | | | Reasoning-Centric | | | AVG |
|---|---|---|---|---|---|---|---|---|---|---|---|
| | | V* | HRBench | InfoVQA | DUDE | SlideVQA | MMLong | MMMU | EMMA | MathVista | |
| *General-ability model* | | | | | | | | | | | |
| GPT-4o | - | 45.0 | 65.0 | 80.7 | 52.7 | 53.1 | 42.3 | 51.9 | 32.7 | 63.4 | 53.5 |
| GPT-4o-mini | - | 50.8 | 48.0 | 83.3 | 46.5 | 48.1 | 28.6 | 45.1 | 27.3 | 60.0 | 48.6 |
| Claude-3.5 | - | - | - | 74.3 | - | - | - | 68.3 | 38.1 | 67.7 | - |
| Qwen2.5-VL-Instruct | 72B | 81.2 | 73.4 | 84.3 | 44.5 | 38.8 | 24.9 | 67 | 38.5 | 74.8 | 58.6 |
| Llava-OV | 7B | 72.8 | 64.7 | 88.3 | 38.1 | 35.0 | 19.5 | 48.8 | 18.3 | 63.2 | 49.9 |
| Qwen2.5-VL-Instruct | 7B | 71.4 | 69.2 | 80.7 | 41.8 | 38.9 | 21.2 | 54.3 | 21.5 | 68.2 | 51.9 |
| MiMO-VL-Instruct | 7B | 78.6 | 70.6 | 84.2 | 43.1 | 48.0 | 27.5 | 51.6 | 26.5 | 78.8 | 56.5 |
| *Capability-Enhanced model* | | | | | | | | | | | |
| VL-Rethinker | 7B | 68.2 | 65.0 | 79.5 | 39.1 | 38.5 | 20.9 | 56.7 | 29.7 | 74.9 | 52.5 |
| R1-VL | 7B | 60.3 | 53.1 | 78.0 | 25.4 | 22.1 | 12.7 | 7.8 | 8.3 | 63.5 | 36.8 |
| OpenVLThinker | 7B | 65.2 | 63.6 | 77.4 | 35.7 | 33.5 | 18.6 | 52.5 | 26.6 | 70.2 | 49.3 |
| Pixel Reasoner | 7B | 84.3 | 72.8 | 86.4 | 44.5 | 42.1 | 22.0 | 50.8 | 19.8 | 65.3 | 54.2 |
| DeepEyes | 7B | 88.9 | 73.1 | 87.7 | 35.2 | 33.0 | 17.5 | 45.2 | 18.1 | 64.9 | 52.1 |
| Docopilot | 8B | 40.1 | 48.3 | 75.0 | 40.7 | 35.7 | 28.8 | 36.6 | 12.1 | 45.0 | 40.3 |
| mPLUG-Owl3 | 7B | 64.5 | 60.1 | 76.3 | 39.5 | 48.8 | 21.0 | 42.9 | 15.0 | 50.7 | 46.5 |
| *Our Models* | | | | | | | | | | | |
| **VL-Scaler** | 7B | 86.6 | 74.2 | 87.0 | 45.1 | 58.3 | 33.1 | 54.8 | 31.3 | 73.8 | **60.5** |
| **VL-Scaler-MiMO** | 7B | 89.0 | 76.0 | 87.8 | 47.6 | 59.6 | 39.5 | 62.7 | 32.1 | 80.3 | **63.8** |

*Table 6.* Composition of the Training Dataset.

| Source Domain | Description | Samples | Key Modalities |
|---|---|---|---|
| **Sci-Graph** (arXiv) | Extracted figures and captions from STEM papers. | 5,902 | Line Plots, Histograms, Schematics |
| **Fin-Sheet** | Financial reports and earning statements. | 3,345 | Dense Tables, Excel Grids |
| **News-Layout** | Newspaper front pages and newsletter scans. | 4,722 | Multi-column Text, Headlines, Insets |
| **Info-Graphic** | Educational and marketing infographics. | 3,738 | Icons, Flowcharts, stylized text |
| **Manuals** | Technical instruction manuals. | 1,968 | Diagrams with callouts, Step-by-step visual instructions |
| **High-Res Scenes** | High-quality natural images and complex scenes. | 8,905 | Dense Natural Scenes |
| **STEM-QAs** | Question and Answer pairs in STEM domains. | 16,271 | Text, Equations, Scientific Notation |
| **Total** | | **44,851** | |

## D. Universal Verbal Program (VP) Prompt

Below is the full system instruction used for the Verification by Verbal Program (VP).

---

**System Prompt for Verification by Verbal Program**

```
You are an impartial AI judge evaluating the semantic similarity between a
Model Response and a Ground Truth Answer.
{question_section}
Ground Truth Answer: {ground_truth}
Model Response: {prediction}
```

**## EVALUATION ALGORITHM**
```
Follow these steps sequentially:
```
**### STEP 1: Identify Answer Type**
```
Determine if the ground truth is:
```
- **Type A:** Numerical (integers, floats, percentages, mathematical
  expressions)
- **Type B:** Simple Phrase (short answer, typically ≤5 words, factual)
- **Type C:** Multi-Choice options (letters such as A, B, C, D, ...)
- **Type D:** Sentence/Paragraph (complete sentence or longer explanation)
```
Write: "Ground Truth Type: [A/B/C/D]"
```
**### STEP 2: Extract Boxed Content**
```
Check if the model response contains \boxed{} command.
Write:
```
- "Boxed content found: [content inside \boxed{}]" OR
- "Boxed content found: None"

**### STEP 3: Apply Type-Specific Scoring**
**#### IF Type A (Numerical) OR Type B (Simple Phrase):**
**3.1** Compare boxed content with ground truth:
- Extract the value/phrase from the model response
- For numerical: Check exact match (account for equivalent representations:
  9.5 million = 9,500,000)
- For simple phrase: Check semantic equivalence
```
Write:
```
- "Model Answer: [extracted content]"
- "Ground truth: [ground truth value]"
- "Match status: [Exact match / Equivalent / No match]"

**3.2** Calculate score:
- If exact/equivalent match: Base score = 1.0
- If no match: Base score = 0.0
- If \boxed{} is missing or improperly used: Deduct 0.1 from base score
  (minimum 0.0)
```
Write: "Type A/B Score: [score]"
```
**### IF Type C (Multi-Choice Options):**
**3.1** Identify the selected option:
- Check boxed content for the letter choice (A, B, C, D, etc.)
- If boxed content contains choice description/content instead of letter,
  match it to the corresponding option
- If boxed content is missing, scan the model response for explicit option
  selection
```
Write:
```
- "Identified option letter: [letter]" OR "Identified option letter: None
  (content only)" OR "Identified option letter: None"
- "Match method: [Direct letter / Matched by content / Not found]"

```
3.2 Ground truth comparison:
Write:
• "Ground truth option: [letter] – [full content of the correct option]"
• "Model's answer: [letter/content as identified in 3.1]"
• "Comparison result: [Correct letter / Correct content only / Incorrect]"
3.3 Scoring rules:
Apply the following score based on the comparison:
• Correct letter in \boxed{} | 1.0
• Correct letter without \boxed{} | 0.9
• Correct letter in \boxed{} + correct content following | 0.7
• Correct content in \boxed{} without letter | 0.3
• Else: 0.0
Write: "Type C Score: [score] – [reason based on table above]"
Note: The scoring prioritizes the option letter, but provides partial
credit when the correct choice content is identified without the letter
#### IF Type D (Sentence/Paragraph):
3.1 Reconstruct reference answer:
Write: "Reference answer sentence: [complete sentence based on question
context and ground truth]"
3.2 Extract implied answer from model:
Write: "Model's implied answer: [sentence/meaning extracted from model
response]"
3.3 Semantic comparison:
Analyze whether both sentences convey the same meaning. Consider:
• Core facts and claims
• Logical equivalence
• Key information preservation
Special case: if the reference answer implies the question is not
answerable, it means the relevant information is not provided in the given
queries. So the model response is correct if it implies similar meanings.
Write: "Analysis: [detailed comparison]"
3.4 Assign semantic similarity score:
• 1.0: Perfect semantic equivalence (same meaning, different wording
  acceptable)
• 0.7-0.9: High similarity with minor differences
• 0.4-0.6: Partial overlap, some key information matches
• 0.1-0.3: Minimal overlap
• 0.0: No semantic overlap or contradictory
Write: "Type C Score: [score]"
### STEP 4: Final Judgment
Provide:
1. "Conclusion: [brief summary of evaluation]"
2. Final score on a new line: \boxed{score}
Important: Use only ONE \boxed{} for the final score.
Now execute the algorithm above step by step.
```

## E. Experiment Details

### E.1. Construction of Eval Set for Validating VP

This experiment is designed to validate the central claims of our Verification by Verbal Program (VP) method. We aim to prove that VP is not only **more accurate** than standard LLM prompting but also significantly **more reliable (low-variance)**. The experiment tests VP's ability to handle semantic ambiguity – where *Rigid Rule* verifiers fail – and subtle inaccuracies –

where standard *LLM Prompting* verifiers often fail.

**Experimental Setup.**

- **Challenge Set:** We manually curated a "VP-Challenge-Set" of 273 (model answer, gold answer) pairs, selected from the both the training set and test set. This set was meticulously labeled by human experts with a binary (Correct/Incorrect) verdict. The set is composed of 144 "Correct" (positive) samples (61 exact matches, 83 semantic rephrasings) and 129 "Incorrect" (negative) samples (79 subtle errors, 50 mixed-fact answers).
- **Verifiers Compared:**
  1. **Rigid Rule:** A verifier using a combination of exact string match, keyword matching, and regular expressions.
  2. **LLM Prompting:** Our base LLM (Qwen2.5-Instruct-14B) prompted with a natural language question (i.e., "Is the model prediction semantically equivalent to the given ground truth answers? output TRUE if equivalent, otherwise FALSE.").
  3. **VP (Ours):** The same LLM judge (Qwen2.5-Instruct-14B) instructed to "execute" our universal Verbal Program (VP) to arrive at a verdict.
- **Metrics:**
  1. **Accuracy (Acc.), F1-Score (F1):** Standard classification metrics comparing the verifier's verdict against the 273 human-annotated ground truth labels.
  2. **Consistency:** To measure reliability, we ran the stochastic verifiers (LLM Prompting and VP) five times for each sample with a non-zero temperature ($T = 0.7$). This metric reports the percentage of samples for which the verifier produced the *exact same verdict* in all 5 runs.

### E.2. Validation of the Perception Verification (PV) Oracle

A core premise of our VL-Scaler framework is the utility of our Perception Verification (PV) mechanism. This mechanism relies on a text-only LLM (Qwen2.5-Instruct-14B), which acts as a "reasoning-oracle proxy," to evaluate whether a VLM's generated perception blocks ($a_p$) are a "sufficient statistic" for answering a given question ($Q$). The proxy's verdict, $R_P \in \{\text{Sufficient}, \text{Insufficient}\}$, is a critical reward signal.

However, this proxy is a high-capability, not an infallible, reasoner (as discussed in Section 2.3). It is susceptible to "fallibility," particularly "false negatives," where it fails to reason from a text block that a human would find sufficient. This experiment is designed to quantify the reliability of our PV oracle by comparing its verdicts against a human-annotated ground truth.

**Data Sampling.** We randomly sampled $N = 979$ data points from our model's rollouts. Each data point consisted of a triplet: (1) the input **Question** ($Q$), (2) the VLM-generated **Perception Blocks** ($a_p$), and (3) the **Golden Answer** ($Y_{gold}$).

**Task Design & Annotation.** We recruited three independent annotators via the Amazon Mechanical Turk (AMT) platform. To ensure high-quality, non-expert annotations, the annotators were given a clear, binary-choice task:

*"Given the 'Question', and the 'Golden Answer', does the provided 'Perception Text' contain all the necessary factual information to logically deduce the 'Golden Answer'? You must not use any external knowledge or information from the (omitted) image. Judge based **only** on the text provided."*

The output for each annotator was a binary label: **Sufficient** or **Insufficient**. We established our human ground truth by taking the majority vote of the three annotators.

## F. Examples

---

**Example Query**

**Question:**
```
Which of the following company logos do not appear in the slides?  Facebook,
AWS, Cargo and Manta.
```
**Ground Truth:**

---

```
AWS
```

## Model Response of VL-Scaler

```
<think>First, I need to understand the nature of these documents.  They
appear to be slides from a presentation about mastering strategic SEO
audits.  The slides seem to discuss background information, avoidances, and
possibly some company logos.  My goal is to determine which company logos
are missing from the slides.  Let's break down the user's request.  They
want me to identify which of the following company logos are not present:
Facebook, AWS, Cargo, and Manta.  I'll start by looking for these logos in
the images provided.</think>
<recognition>Scanning through the images, I notice various company logos,
but I need to carefully check for Facebook, AWS, Cargo, and Manta.  After
a thorough examination, I can see that Facebook, Cargo, and Manta are
indeed present in the images.  However, AWS does not appear anywhere in the
slides.</recognition>
<think>Great, I've found that AWS is the only logo among the four that is
not present in the slides.  Now, I need to confirm my findings.  Let me
double-check the images one more time to ensure accuracy.</think>
<recognition>After another careful scan, I confirm that Facebook, Cargo, and
Manta are all present in the images.  AWS is still missing.  This means that
AWS is the correct answer to the user's request.</recognition>
<think>So, the company logo that does not appear in the slides is AWS. This
matches the solution I was given.  I'm confident in this conclusion because
I've double-checked the images multiple times.  Before finalizing, I should
reflect on whether I missed anything.  Wait, let me double-check again just
to be sure.</think>
<recognition>After the final scan, I reaffirm that Facebook, Cargo, and
Manta are all present, while AWS is missing.  This confirms that AWS is the
correct answer.</recognition>
<think>Alright, everything checks out.  The company logo that does not
appear in the slides is AWS. This is the final answer.  I'm satisfied with
this conclusion because it aligns with the evidence I've gathered from the
images.</think>
<answer>   The company logo that does not appear in the slides is AWS, as it
is the only one among the options Facebook, AWS, Cargo, and Manta that is
missing from the images. </answer>
```

## Model Response of Baseline GRPO

```
Let's analyze the company logos shown in the images step by step:
1.  Image 1 (Background Information):
-Petco
-Disney Interactive Studios
-NBC Universal
-Mashable
-GSN
-UGG Australia
-Beats by Dr.  Dre
-Chamber of Commerce
…
```

```
2.  Image 2 (Shiny Object Distractions):
-Addthis
-AIM
-AIM (alt)
-Apple
-Bebo
…
Now, let's compare the logos from Image 1 and Image 2 to the list of logos
provided in the question (Facebook, AWS, Cargo, and Manta):
-Facebook is present in Image 2.
-AWS is not present in either image.
-Cargo is present in Image 2.
Based on the analysis, the company logos that do not appear in the slides
are AWS and Manta.
```

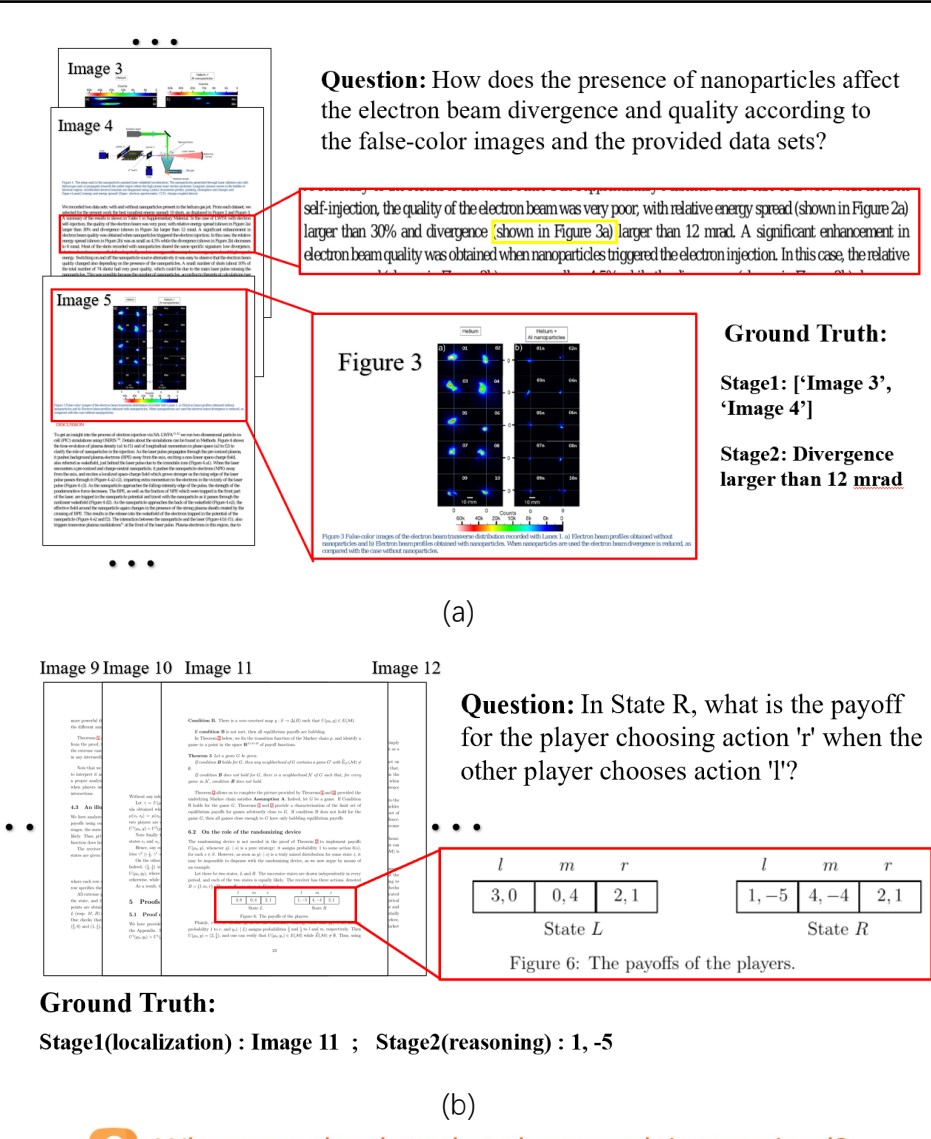

(a)

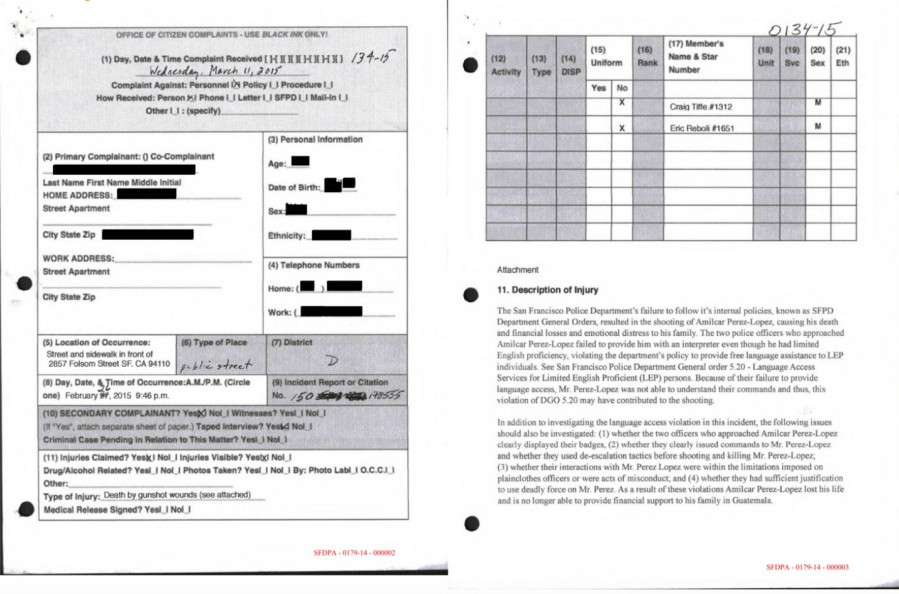

(b)

(20)

*Figure 5.* Example Queries of Collected Modality-Rich Dataset.