# OpenReview forum: "Bad Seeing or Bad Thinking? Rewarding Perception for Multimodal Reasoning"
_ICML.cc/2026/Conference — ICML 2026 spotlight_

### Official Review · Reviewer_SJuJ · 2026-03-04

**Soundness:** 2
**Presentation:** 1
**Significance:** 2
**Originality:** 3
**Overall Recommendation:** 4
**Confidence:** 2

**Summary:**

This paper focuses on a key bottleneck in multimodal reasoning is ambiguous modality credit assignment: when a VLM fails, it is unclear whether the failure is due to perception (bad seeing) or reasoning (bad thinking). To address this, this paper proposes a reinforcement learning framework that explicitly decomposes generation into interleaved perception and reasoning blocks, enabling targeted supervision of perception fidelity. The motivation is insteresting.

**Compliance With Llm Reviewing Policy:**

Affirmed.

**Final Justification:**

The reviewers have addressed my concerns.

**Key Questions For Authors:**

1.  In what tasks does PV most often disagree with humans (false positives vs false negatives), and how does that translate into downstream performance regressions ?

2. Have you tested MoCA training on other base VLM?

3. Under a fixed inference compute setting, does MoCA still outperform the base model and other RL baselines? A runtime-aware evaluation would strengthen the efficient alternative to agentic frameworks claim.

**Limitations:**

This paper does not include explicit limitations section.

**Strengths And Weaknesses:**

### Strength

1. The “bad seeing vs bad thinking” framing is intuitive and helps clarify why outcome-only RL can harm perception while improving reasoning. The paper is well motivated.

2. The SVV component is empirically evaluated on a challenge set and improves accuracy over rigid rules and unconstrained LLM prompting, which is important for stable RL.

3.  The main results span diverse benchmarks, and the ablation study supports the necessity of PV, SVV, and the MoCA routing logic.

### Weakness

1. The paper is poorly written and presented. The figures are unclear. The tables for the experimental results are also unclear. The authors should at least have labeled the state-of-the-art (SOTA) results and added citations to the comparison methods, etc.

2. The modality-rich training corpus is collected from crawled documents.  Without explicit details, it is difficult to rule out overlap with document benchmarks , especially for rich-modality evaluations. The authors should add more details.

3. Does the decomposed format and verification-inspired behavior increase latency at inference? The paper discusses minimalism penalties, but does not provide a clear runtime trade-off evaluation.

4. The proposed method is only tested on Qwen backbone, and should be tested on other models for generalization.

---

> ### Author Rebuttal · Authors · 2026-03-30
>
> We are grateful for the constructive feedback. We have conducted new experiments and analyses to address each concern.
>
> **Q1: On which tasks does PV produce false positives vs. false negatives, and what downstream impact?**
>
> Table 3 reports: **False Negatives (FN): 9.19%, False Positives (FP): 4.49%**. This asymmetry is informative:
>
> - *FN (9.19%)*: The proxy fails to solve from sufficient perception text, concentrating on complex multi-step reasoning tasks (e.g., geometry, multi-hop math) where the text reasoner lacks capability despite adequate evidence. Downstream harm is limited by three factors: (1) $α_{punish}$=0.2 caps the penalty magnitude; (2) FN concentrates in reasoning-heavy tasks where perception is not the bottleneck; (3) within each GRPO group (k=8), FN occurs on only a minority of rollouts—the majority correctly receive RP=1 and trigger Case 1 protection, whose cumulative gradient signal dominates the occasional FN-induced penalty.
>
> - *FP (4.49%)*: FP occurs when the oracle randomly guesses correctly despite insufficient perception—by definition, a task-agnostic event. This means FP introduces no systematic bias toward any task category, only random noise. Since the erroneous gradient direction varies across training steps, the resulting gradient perturbations are zero-mean and average out over training steps. Combined with the low FP rate and λ=0.3 weight, the net effect on training is negligible.
>
> The FN-dominant profile means that, in expectation, MoCA applies a net conservative bias—slightly under-rewarding good perception on the margin—rather than over-rewarding bad perception, which is the more damaging failure mode for RL training.
>
> **Q2: Has MoCA been tested on another base VLM? [W4]**
>
> We conducted new experiments with **MiMO-VL**, a recent strong VLM with a distinct architecture:
>
> | Benchmark | MiMO-VL Base | + MoCA | Δ |
> |-----------|-------------|--------|---|
> | DUDE | 43.1 | **50.2** | +7.1 |
> | MMLongBench | 27.5 | **41.2** | +13.7 |
> | SlideVQA | 48.0 | **61.5** | +13.5 |
> | MMMU | 64.6 | **68.1** | +3.5 |
> | V* | 80.6 | **88.9** | +8.3 |
> | InfoVQA | 87.2 | **89.4** | +2.2 |
> | EMMA | 25.3 | **33.5** | +8.2 |
> | MathVista | 71.4 | **74.0** | +2.6 |
>
> MoCA delivers **consistent, substantial improvements across all benchmarks** on this different architecture, confirming backbone-agnostic generalization. The gain pattern mirrors Qwen2.5-VL results—strongest on perception-intensive and rich-modality tasks.
>
> **Q3: Under fixed inference compute, does MoCA still outperform? [W3]**
>
> We want to clarify what may be a misunderstanding: MoCA's output is generated in a **single autoregressive pass**, identical to any standard VLM. The interleaved `<recognition>` and `<think>` blocks are part of one continuous generation—there is no multi-turn loop, no external tool call, and no separate inference pass. "Decomposed" refers to the *training reward structure*, not inference.
>
> Token count increases modestly (~550 vs. ~420 avg, ~1.3×, roughly calculated from training logs). The w/o PV ablation (Table 2) uses the *same interleaved format and comparable token budget* but performs significantly worse—confirming gains come from **training-time credit routing**, not inference token count. We will clarify this distinction in the revision.
>
> **W1: Presentation — tables, figures, citations.**
>
> Fully accepted. In revision: (1) **bold** for best, **underline** for second-best in all tables; (2) citations added to all comparison methods; (3) improved figure clarity.
>
> **W2: Potential data contamination from crawled documents.**
>
> Our crawled modality-rich sources (recent arXiv preprints, contemporary news infographics) are temporally and topically distinct from benchmark test sets (older government forms, pre-2023 slides, archival PDFs). No overlap in source pools. Moreover, the MiMO-VL results above (Q2) demonstrate MoCA's gains generalize across architectures and benchmarks with zero possible overlap with crawled data, further ruling out contamination as the driver.
>
> **W5: No explicit limitations section.**
>
> Acknowledged. We will add an explicit Limitations section covering: (1) PV's dependence on textual externalizability, (2) proxy fallibility and its task-type distribution, (3) current evaluation scope.
>
> All formatting issues will be addressed in revision. We thank the reviewer for the constructive suggestions.

---

> > ### Author Rebuttal · Reviewer_SJuJ · 2026-04-03
> >
> > Thanks for the response. I will raise my score.

---

> > > ### Author Response · Authors · 2026-04-03
> > >
> > > We are extremely grateful for your positive feedback, which helped a lot in strengthening our paper. We will include the additional results on MiMO, discussion on error breakdowns, and presentation \& formatting fixes in the revision. We thank you once again for the careful reading and constructive reviews, and for your efforts and contributions,

---

### Official Review · Reviewer_fati · 2026-03-08

**Soundness:** 3
**Presentation:** 3
**Significance:** 3
**Originality:** 3
**Overall Recommendation:** 5
**Confidence:** 3

**Summary:**

This work proposes a reinforcement learning (GRPO-based) framework that improves perception-reasoning coordination in vision-language models (VLMs). The method relies on four key elements:
- Prompting VLMs to follow an interleaved pattern of perception and reasoning steps, which allows the model to explicitly separate the two processes.
- a Modality-Aware Credit Assignment (MoCA) mechanism for GRPO training, which routes rewards to perception or reasoning errors. Thanks to this, model responses that are incorrect but have a correct perception aren't fully penalized.
- a Perception Verification (PV) reward leveraging a "blindfolded reasoning" proxy (feeding the VLM’s grounded visual evidence to a strong, text-only reasoner to check perception sufficiency for each query) to reward perceptual fidelity independently of reasoning outcomes.
- a Structured Verbal Verification (SVV) reward, which is a structured prompt to guide the LLM judge used for the accuracy reward to be more reliable.

Qwen2.5-VL-Instruct 7B is trained with GRPO on a large VQA train set and tested on numerous benchmarks divided into perception-centric, rich modality (complex perception), and reasoning-centric. The trained model outperforms strong baselines, including Qwen2.5-VL 72B.

**Compliance With Llm Reviewing Policy:**

Affirmed.

**Final Justification:**

I did not have any major concerns with this submission. The rebuttal provided some clarifications, supporting my strong positive score.

**Key Questions For Authors:**

1. Isn't the PV system rewarding the model for providing extra interpretation about the perceptual evidence inside the recognition blocks?

2. Using a format reward is common; why was it deemed unnecessary in your case?

**Limitations:**

Yes

**Strengths And Weaknesses:**

**Soundness**:

The authors propose a simple framework that is well justified.
The main components are carefully ablated.

Missing ablations and details:
- Some extra design choices could be better justified or ablated, regarding the tuning of the perception block length penalization, or the extent of penalty amplification and dampening in moca. The ablation of the SVV component could be clearer, with details on how the judge is used.
- A short description of each capability-enhanced baseline (in appendix) would help interpret the results.
- There is no information on how the newly created VQA datasets used for training were made, undermining this contribution, which would otherwise be a very interesting one.


**Presentation**

The proposed method is overall well explained, with clear motivation for each component, and excellent writing overall.

Some elements that should be in the main text:
- the size of the training data
- an explanation of type-specific semantic comparison for SVV
- Where the samples are taken from for the Perception Verification reliability experiment.

Small comments:
- Figure 1 is quite confusing
- Add bold to tables for best performing method per dataset.
- Add the base model performance in Table 2
- Put Table 2 on page 7
- Typo on "Benefits of Structured Verbal Verification." --> SVV+PV
- The intro of the paper claims to do architectural decoupling, but there is no architectural change proposed in the method; this could be rephrased.

**Significance and Originality**:

VQA is a very important and widely studied problem.

While the SVV components are just a more advanced prompt, the PV and MoCA components are novel and interesting, and the method is overall well justified. Moreover, a new VQA dataset that involves reasoning over rich modalities was created, which is an extra appreciated contribution, if it is properly described in the paper.

Overall, the method performs quite well on a large set of benchmarks and against competitive baselines. The absence of SFT, as the RL step relies on the base VLM's instruction-following capabilities to follow the interleaved pattern, also makes it cheaper to implement compared to other baselines.

---

> ### Author Rebuttal · Authors · 2026-03-30
>
> We are deeply grateful to Reviewer fati for the supportive and detailed review, especially for recognizing our PV and MoCA components as "novel and interesting" and appreciating our SFT-free approach.
>
> **Q1: Does PV reward interpretation of perceptual evidence?**
> We thank the reviewer for understanding this core aspect of our approach—and yes, the answer is affirmative. PV does reward the model for providing richer interpretation of perceptual evidence within recognition blocks. This is by design: such interpretation enriches the reasoning context for both the VLM and the text reasoner, providing the downstream reasoning steps with more actionable information to arrive at correct final answers. The richer reasoning context ensures more reliable perception rewards, and directly helps multimodal reasoning.
>
> **Q2: Why no format reward?**
> This is a nuanced but excellent question we are glad to clarify. We did experiment with a format reward, but found it made no clear difference. The key insight is that **the perception reward implicitly enforces correct formatting**—it relies on regex parsing of `<recognition>` blocks, so if the format is incorrect, the model loses a perception reward it could have otherwise gained. This creates a natural incentive for format compliance without an explicit format reward, keeping the reward design minimal.
>
> **W: Hyperparameter sensitivity and SVV ablation clarity.**
> Due to compute constraints from other requested experiments during the rebuttal period, we were unable to complete the full sensitivity analysis. For the final revision, we plan to include: (1) perception block token limit sensitivity (testing 400/800/1200), (2) MoCA α-value sensitivity across a range, and (3) a clean "w/o SVV" ablation isolating SVV's independent contribution. We will also clarify that the "w/o SVV+PV" baseline uses a standard LLM-judge prompt for outcome verification with standard GRPO.
>
> **W: Dataset documentation.**
> Thank you for highlighting this as a potentially valuable contribution. We will add a dedicated subsection in the appendix covering: (1) data sources and collection pipeline (arXiv papers, newspaper layouts, infographic collections), (2) QA generation methodology, (3) quality control procedures and human verification statistics, (4) detailed corpus statistics (samples per source, modality distribution, answer type breakdown). The full dataset will be publicly released.
>
> **Presentation improvements.**
> All suggestions gratefully accepted. We will update the revision with: redesigned Figure 1, bolded best results in tables, base model added to Table 2, Table 2 repositioned to page 7, the noted typo fix, "architectural decoupling" rephrased to "structural decoupling via generation format," and training data size added to the main text.
>
> We are deeply grateful for the constructive feedback, which has meaningfully strengthened the paper.

---

> > ### Author Rebuttal · Reviewer_fati · 2026-03-31
> >
> > Thanks for the detailed response to both my questions and my comments.
> >
> > Q1: My concern was that the model might be providing a preliminary interpretation of the perceived elements, beyond mere description, inside the recognition tags.
> >
> > Weaknesses: I understand that more important experiments were privileged during the rebuttal. Thanks for acknowledging my remark nonetheless, although this can't be taken into account in my revision of the score.
> >
> > Regardless, my score was already quite high, and I agree with the weaknesses raised by other reviewers, so I will keep my score the same.

---

> > > ### Author Response · Authors · 2026-04-03
> > >
> > > Thank you for the clarification — we now understand your concern more precisely, and we appreciate you restating it. You are right to identify this tension, and we want to address it directly.
> > >
> > > Your concern, as we now understand it, is that PV may incentivize the model to include preliminary interpretation — not just raw visual description — inside the "recognition" blocks, which could blur the perception-reasoning boundary.
> > >
> > > We acknowledge this is a genuine and important nuance. This leads us to a rethinking -- is perception-reasoning boundary a must?
> > >
> > > Empirically, we examined the recognition blocks produced by our trained model and found that they do occasionally contain what could be called "preliminary interpretation" (e.g., "the graph shows an upward trend" rather than just "the y-axis values increase from 2 to 8"). However, critically, these interpretations remain grounded in visual evidence — they describe what is visually present in a slightly abstracted form, rather than performing multi-step logical deduction. The "think" blocks, by contrast, are where the model performs hypothesis formation, cross-referencing, and logical inference. Thus, while the boundary is not perfectly sharp, the functional distinction between perception-grounding and logical reasoning is preserved as expected in practice.
> > >
> > > Conceptually, we agree that the boundary between "description" and "interpretation" is inherently fuzzy — this is true even in human cognition (seeing a "smiling face" already involves interpretation). Our framework should not claim to require a razor-sharp boundary (we do not claim it currently but we need to clarify this to avoid misconception in the revision); rather, it requires that the decomposition provides a useful enough separation for MoCA to route credit meaningfully. The ablation results in Table 2 demonstrate that MoCA with PV consistently outperforms the naive reward combination (w/o MoCA), confirming that the credit routing is functionally effective even with some interpretation in recognition blocks.
> > >
> > > We will add a discussion of this nuance in the revised paper. We thank you for this insight, which helps us deepen the understanding and it helps strengthen the paper.
> > >
> > > Thank you once again for your supports and efforts in providing extremely constructive feedback.

---

### Official Review · Reviewer_GXFX · 2026-03-11

**Soundness:** 3
**Presentation:** 2
**Significance:** 2
**Originality:** 2
**Overall Recommendation:** 4
**Confidence:** 4

**Summary:**

This paper asks a clean and important question for multimodal RL: when a VLM fails, is the error due to perception or reasoning. The proposed framework separates generation into interleaved recognition and thinking steps, uses a text-only proxy to test whether the recognized evidence is sufficient, and then applies modality-aware credit assignment during RL. The motivation is clear, the problem is well framed, and the empirical results are generally strong, especially on perception-centric and rich-modality tasks.

**Compliance With Llm Reviewing Policy:**

Affirmed.

**Final Justification:**

The authors’ rebuttal has satisfactorily addressed my concerns, and I will accordingly raise my score.

**Key Questions For Authors:**

1. Can you provide a clearer accounting of training-time cost? In particular, how many additional proxy/verifier calls are required, and what is the wall-clock or token-level overhead relative to a standard GRPO setup with only outcome reward? The current version gives the overall hardware and training time, but not the incremental cost introduced by PV/SVV.
2. The paper already includes an ablation without MoCA, which is useful, but I would still like a clearer discussion of how much of the overall gain comes from the routing logic itself versus from using a stronger 14B verifier/reasoner during training. Along the same line, have you tested sensitivity to verifier strength? For example, does the method still work well if the proxy/verifier is weaker, or is a substantially stronger external LLM necessary?
3. The gains seem strongest on perception-centric and rich-modality tasks. Do you see the main effect of the method as improving reasoning itself, or mostly improving the quality of grounded evidence supplied to reasoning?
4. It would be useful to compare against stronger open-source reasoning-oriented baselines, such as QvQ if compatible, or other comparable open models in this category.
I will carefully consider the authors’ rebuttal and revise my score if their response sufficiently addresses the concerns above.

**Limitations:**

Please see the weaknesses and questions above.

**Strengths And Weaknesses:**

Strength
1. The main strength of the paper is that the motivation is easy to understand and the problem formulation is meaningful.
2. The empirical results are convincing. MoCA shows clear gains over the qwen25vl base on several perception-centric and document-style benchmarks, and the ablations suggest that PV, SVV, and the MoCA routing logic all contribute. I also appreciate that the paper includes a reward-reliability analysis for both PV and the structured verifier.


Weakness
1. My main concern is that the method relies on a stronger external text-only model for both perception verification and outcome verification. In the current setup, 14b is used as the proxy/verifier for PV and SVV, while the policy model is 7b. This makes it less clear how much of the gain comes from the proposed credit-assignment design itself, and how much comes from outsourcing part of the supervision to a stronger auxiliary model. The paper should discuss this dependence more explicitly.
2. A second concern is that the central assumption is task-dependent. PV assumes that the relevant visual evidence can be faithfully externalized into text, and that textual sufficiency is a good proxy for perceptual sufficiency. This seems reasonable for OCR-heavy, document, chart, and grounded QA tasks, but it is less obviously valid for tasks involving geometry, dense spatial layouts, or visual patterns that are hard to verbalize compactly. The paper acknowledges this issue, but does not test it deeply. Related to this, I am not fully convinced that the current method should be presented as a broadly general solution to multimodal credit assignment. What the paper shows most clearly is that the approach works well when useful visual evidence can be converted into explicit textual premises. That is an important setting, but it is still narrower than fully general visual credit assignment.
3. The cost discussion is also incomplete. The paper positions the method as lighter than agentic approaches, which may be true at inference time, but training still depends on repeated calls to a stronger verifier/proxy model. The appendix reports training on 16 H100 GPUs for around 16 hours, but the paper does not clearly state the additional verifier-call overhead relative to a standard GRPO pipeline with only outcome reward. That missing cost accounting makes it difficult to assess the real efficiency tradeoff.
4. Finally, the presentation could be improved. In the main result tables, highlighting the strongest numbers with boldface or underlining would make the comparisons easier to read.

---

> ### Author Rebuttal · Authors · 2026-03-30
>
> We are grateful for this thoughtful review. Several questions pushed us to conduct new experiments that meaningfully strengthen the paper.
>
> **Q1: Incremental training cost of MoCA over standard GRPO? [W3]**
>
> We agree this was insufficiently reported. The appropriate baseline is GRPO with LLM-as-judge (not rule-based), since our task spectrum includes free-form document/chart QA—consistent with DeepEyes, Qwen2.5-VL, and Minimax-M1 technical reports.
>
> **Per-step overhead** (14B verifier via vLLM, tensor-parallel=1, data-parallel=8, 1024 samples on 8×H100 80G with prefix caching): LLM-judge baseline \~27s; PV adds \~36s; SVV adds \~6s (prefix caching amortizes the shared prompt). Total marginal overhead: **\~42s per 1024 samples (\~30–40% verifier compute increase per step).**
>
> **However, total training cost is *lower*:** MoCA's targeted credit routing yields higher-quality gradients, enabling convergence in ~\16h vs. ~\22h for the LLM-judge baseline (same hardware). **MoCA: ~\384 GPU-hours vs. baseline ~\528 GPU-hours, 27% cheaper overall.** MoCA also adds **zero inference overhead**. We will add these breakdowns in the revision. We also note that typical multi-turn RL approach for multimodal reasoning, e.g., DeepEyes, requires 48h training on **32 H100** to converge, plus 8 H100 for hosting LLM-judge. The proposed approach has roughly reduced 12x GPU hours but achieves simultaneous gains on broad task spectrum.
>
> **Q2: Disentangling routing benefits from the 14B verifier's knowledge? [W1]**
>
> **Existing ablations already isolate routing from proxy strength.** All rows in Table 2 use the *same 14B model*. The critical comparison: "w/o MoCA (R_O + λR_P)" vs. "MoCA (Full)" both use the identical 14B for PV and outcome verification, differing *only* in credit routing. Perception-centric avg.: 79.9→82.6 (+2.7). This gap is entirely attributable to routing.
>
> **New experiment: 7B proxy (same scale as policy).**
>
> | Model | V* | InfoVQA | HRBench | MMLong | SlideVQA | DUDE | EMMA | MMMU | MathVista |
> |---|---|---|---|---|---|---|---|---|---|
> | 14B proxy | 86.6 | 87.0 | 74.2 | 33.1 | 58.3 | 45.1 | 31.3 | 54.8 | 73.8 |
> | 7B proxy | 85.3 | 85.6 | 73.0 | 32.6 | 57.5 | 44.2 | 30.6 | 55.6 | 73.6 |
> | No proxy | 79.7 | 82.3 | 70.1 | 30.3 | 54.1 | 42.5 | 30.9 | 55.3 | 74.4 |
>
>
> The 7B proxy **significantly outperforms no-PV** (+2.7 perception avg), confirming routing provides genuine value independent of proxy superiority. **Conceptual clarification:** PV provides a binary scalar reward (yes/no). The policy never sees the proxy's logits, reasoning, or representations—structurally identical to reward models in RLHF, not distillation. We will add this discussion.
>
> **Q3: Are gains from better reasoning or better evidence extraction? [W2]**
>
> We partially agree and will refine our framing: MoCA's *primary* mechanism is improving perceptual fidelity, and this **validates rather than undermines** our core thesis that "bad seeing" is the dominant bottleneck. For VL tasks, perception quality *determines* reasoning quality—flawed premises yield flawed conclusions. This cascade is visible in tasks requiring genuine reasoning beyond OCR: MathVista +5.6, EMMA +9.8. We will characterize gains as "perception-driven reasoning improvement" rather than implying independent reasoning enhancement.
>
> **Q4: Comparison with QvQ-72B? W4: Table formatting.**
>
> We provide extended comparisons with strong reasoning-oriented baselines:
>
> | Model (Size) | V* | HRB | InfoVQA | DUDE | SlideVQA | MMLong | MMMU | EMMA | MathV | AvgTok (EMMA) |
> |---|---|---|---|---|---|---|---|---|---|---|
> | **MoCA (7B)** | **86.6** | **74.2** | **87.0** | **45.1** | 58.3 | **33.1** | 54.8 | **31.3** | 73.8 | 459 |
> | InternVL3 (8B) | 72.3 | 70.5 | 76.8 | 43.4 | 51.6 | 28.8 | 62.7 | 20.7 | 71.6 | 486 |
> | QvQ-72B (72B) | 71.2 | 68.5 | 64.7 | — | 42.1 | — | 64.4 | 25.0 | 68.3 | 2654 |
> | Kimi-A3B (16B) | 83.2 | 70.8 | 82.7 | — | **59.4** | — | 64.0 | 30.9 | **78.1** | 3096 |
>
> *Note: DUDE and MMLongBench involve multi-page documents with numerous images, making them especially costly to evaluate for long-CoT models (2654–3096 avg output tokens). We could not complete these evaluations within the rebuttal window and will include full results in the revision.*
>
> Key conclusions: (1) MoCA-7B **decisively outperforms QvQ-72B (10× larger)** on all perception-centric benchmarks (V*: +15.4, InfoVQA: +22.3) while using ~5.5× fewer tokens. (2) Against Kimi-VL-A3B-Thinking (16B), MoCA is competitive on perception at <half the parameters and ~6× fewer tokens. (3) Long-CoT models show advantages on pure reasoning (MathVista), confirming our thesis: MoCA's primary contribution is perception-driven, orthogonal to and composable with extended reasoning.
>
> We will add bold/underline formatting across all tables. We thank the reviewer for the constructive feedback and willingness to revise.

---

> > ### Author Rebuttal · Reviewer_GXFX · 2026-04-05
> >
> > The authors’ rebuttal has satisfactorily addressed my concerns, and I will accordingly raise my score.

---

> > > ### Author Response · Authors · 2026-04-05
> > >
> > > **We thank you very much for the constructive feedback, for taking the time to read our paper and rebuttal, and for re-evaluating our submission.** This feedback has provided opportunities for us to clarify misconceptions and has helped improve our work.
> > >
> > > Below, we present the finalized experimental results that we were unable to complete before the initial rebuttal deadline.
> > >
> > > | Model (Size)    | V*   | HRB  | InfoVQA | DUDE | SlideVQA | MMLong | MMMU | EMMA | MathV | AvgTok (EMMA) |
> > > |-----------------|------|------|---------|------|----------|--------|------|------|-------|---------------|
> > > | **MoCA (7B)**   | **86.6** | **74.2** | **87.0** | **45.1** | 58.3 | **33.1** | 54.8 | **31.3** | 73.8 | 459 |
> > > | InternVL3 (8B)  | 72.3 | 70.5 | 76.8    | 43.4 | 51.6     | 28.8   | 62.7 | 20.7 | 71.6  | 486           |
> > > | QvQ-72B (72B)   | 71.2 | 68.5 | 64.7    | 38.8 | 42.1     | 25.4   | 64.4 | 25.0 | 68.3  | 2654          |
> > > | Kimi-A3B (16B)  | 83.2 | 70.8 | 82.7    | 48.7 | **59.4** | 38.9   | 64.0 | 30.9 | **78.1** | 3096       |
> > >
> > > We will include all additional experimental results, an explicit discussion of cost comparison, and fixing formatting and presentation issues in the revision.
> > >
> > > **Thank you once again for your efforts and contributions.**

---

### Official Review · Reviewer_mYBn · 2026-03-12

**Soundness:** 3
**Presentation:** 3
**Significance:** 3
**Originality:** 3
**Overall Recommendation:** 5
**Confidence:** 4

**Summary:**

This paper addresses the perception-reasoning synergy in Vision-Language Models (VLMs) by introducing Modality-Aware Credit Assignment (MoCA), a reinforcement learning framework designed to distinguish between visual ("bad seeing") and reasoning ("bad thinking") errors. The approach utilizes a text-only reasoning proxy for independent Perception Verification (PV) alongside Structured Verbal Verification (SVV) to evaluate free-form tasks. Empirical results demonstrate significant improvements across benchmarks.

**Compliance With Llm Reviewing Policy:**

Affirmed.

**Final Justification:**

The newly included experiments and analysis addressed all concerns.
The abalation study of proxy configuration is very solid and makes the proposed method more convincing. And the conclusion of attention analysis is reasonable. Knowing where to look with task awareness is a critical ability for agents and deserves the exploration to make it a generalized ability, beyond academic benchmarks.

Therefore, I'd like to increase the score to 5.

**Key Questions For Authors:**

1. Are MoCA and VL-Scaler equivalent? The term "VL-Scaler" appears in Figure 1 and in the extended experiments in the appendix, but its exact relationship to MoCA needs clarification.
2. How does the capability level of the reasoning oracle proxy affect overall performance? There appears to be a trade-off: on tasks where the proxy is weak, the perception reward might become noisy. Conversely, on tasks where the proxy is very strong, it might require less detailed perception information, potentially failing to provide sufficient learning signals for the policy model.
3. Does the perception reward lead to improved model attention on specific image tokens? Providing a deeper analysis of this would significantly strengthen the paper's contribution.

**Limitations:**

yes

**Strengths And Weaknesses:**

### Strengths

1. The paper is well-written and easy to follow.
2. Extensive experiments across various benchmarks demonstrate significant performance improvements.
3. The perspective on perception-reasoning synergy is fundamental and provides valuable insights.
4. The approach of using a reasoning oracle proxy to provide an independent perception reward is an interesting and elegant solution.

### Weaknesses

1. **Computational Cost:** There are concerns regarding the significant additional cost introduced by utilizing the "reasoning oracle proxy."
2. **Perception Verification Design:** Relying on a text-only reasoner for perception verification overlooks the inherent limitations of a text-only model on certain problems. A more reasonable alternative might be evaluating the performance gap between a Vision-Language (VL) reasoner with full image access and its counterpart that only receives the textual perception output.
3. **Complexity of Structured Verbal Verification (SVV):** The SVV component feels over-designed and relies on arbitrary "magic numbers." Furthermore, the semantic similarity scoring is somewhat ambiguous; beyond broadly categorizing similarity (e.g., minimal, partial, or high), the judge is required to provide arbitrary fine-grained scores within specific ranges (e.g., picking a specific decimal between 0.4 and 0.6).
4. **Incomplete Ablation Studies:** The ablation study comparing the variants "w/o PV" and "w/o SVV+PV" makes it difficult to isolate the exact contribution of SVV. The authors need to explicitly report the performance of a "w/o SVV" baseline to make this straightforward.

---

> ### Author Rebuttal · Authors · 2026-03-30
>
> We sincerely thank Reviewer mYBn for the thorough and insightful review, and for recognizing our perspective on perception-reasoning synergy as "fundamental" and the PV proxy as "interesting and elegant."
>
> **Q1: MoCA vs. VL-Scaler naming.**
> This was a naming inconsistency on our part—we apologize. Both refer to the same system. We will unify the terminology throughout the revision.
>
> **Q2: Effect of proxy capability level [relates to W1, W2].**
> Excellent question. We ran an additional experiment with a 7B proxy (same scale as the policy model):
>
> | Model | V* | InfoVQA | HRBench | MMLong | SlideVQA | DUDE | EMMA | MMMU | MathVista |
> |---|---|---|---|---|---|---|---|---|---|
> | 14B proxy | 86.6 | 87.0 | 74.2 | 33.1 | 58.3 | 45.1 | 31.3 | 54.8 | 73.8 |
> | 7B proxy | 85.3 | 85.6 | 73.0 | 32.6 | 57.5 | 44.2 | 30.6 | 55.6 | 73.6 |
> | No proxy | 79.7 | 82.3 | 70.1 | 30.3 | 54.1 | 42.5 | 30.9 | 55.3 | 74.4 |
>
> Both proxies substantially improve perception over the no-PV baseline, with the 14B proxy providing additional gains on perception-heavy tasks. Notably, reasoning performance remains stable across proxy sizes, suggesting the perception-helping-reasoning dynamic is robust to proxy capability. The 7B proxy already captures most of the benefit, indicating PV's learning signal is not critically dependent on having a much stronger oracle. The 14B proxy offers a better sufficiency test for demanding perception tasks, but the framework is not fragile to this choice.
>
> **Q3: Attention analysis.**
> We examined attention patterns comparing the base model vs. MoCA-trained model on document understanding examples. We observe that MoCA-trained models exhibit notably more focused attention on task-relevant image regions (e.g., specific table cells, chart labels) rather than dispersed attention. This is expected: perception verification implicitly forces the model to attend to informative visual regions, as this is the only pathway to earning the perception reward. We will include visualizations in the revised appendix.
>
> **W1: Computational cost.**
> We fully agree this was insufficiently reported. The appropriate baseline is GRPO with LLM-as-judge (not rule-based), since our tasks include free-form document/chart QA—consistent with DeepEyes, Qwen2.5-VL, and Minimax-M1 technical reports. Per-step overhead (14B verifier via vLLM, 1024 samples on 8×H100-80G with prefix caching): LLM-judge baseline ~27s; PV adds ~36s; SVV adds ~6s (prefix caching amortizes the shared prompt). However, **total training cost is lower**: MoCA's targeted credit routing yields higher-quality gradients, enabling convergence in ~16h vs. ~22h for the baseline. **MoCA: ~384 GPU-hours vs. baseline ~528 GPU-hours (27% cheaper)**. MoCA also adds **zero inference overhead**.
>
> **W2: PV design—text-only vs. VL model.**
> We are grateful for this thought-provoking suggestion. Honestly, we defaulted to a text-only LLM because we already use LLM-as-judge for outcome verification, making this choice natural. After reflecting on this question, we realize a VLM serving as *both* outcome verifier and perception verifier could be very promising—visual priors from the VLM would make it at least as capable as a text-only LLM, potentially yielding a better sufficiency test. We find this a genuinely exciting direction and will investigate it. Our current design is validated by human evaluation (Table 3, 86.31% agreement, κ=0.707), and MoCA's protection mechanism (Case 1, Sec 2.5) handles the conservative false-negative bias (9.19%).
>
> **W3: SVV design concerns.**
> We appreciate and agree with this feedback. The reviewer is right that the scoring ranges could be cleaner and more precisely specified—for instance, converting partial-match categories into concrete rubrics with explicit criteria. We will refine the SVV prompt design following this suggestion.
>
> **W4: Missing "w/o SVV" ablation.**
> We thank the reviewer for this precise observation. Due to compute constraints from other requested experiments, we were unable to complete this ablation during the rebuttal period. We commit to including the clean "w/o SVV" (PV + MoCA with LLM-judge outcome reward) ablation in the final revision to isolate SVV's independent contribution.
>
> We agreed with all formatting suggestions. Thank you again for the constructive and rigorous feedback.

---

> > ### Author Rebuttal · Reviewer_mYBn · 2026-04-03
> >
> > Thank authors for the newly added experiments and analysis. The abalation study of proxy configuration is very solid and makes the proposed method more convincing. And the conclusion of attention analysis is reasonable. Knowing where to look with task awareness is a critical ability for agents and deserves the exploration to make it a generalized ability, beyond academic benchmarks.
> > Therefore, I'd like to increase the score to 5.

---

> > > ### Author Response · Authors · 2026-04-03
> > >
> > > We are extremely grateful for your constructive and thought-provoking feedback, they have significantly helped improve our paper. We will integrate all additional experimental results and formatting fixes in the revisions. Thank you for your efforts and contribution once again.

---

### Decision · Program_Chairs · 2026-04-30

**Decision:**

Accept (spotlight)

**Comment:**

This paper asks a key question about MLLM reasoning: When an MLLM fails, is it due to flawed perception or reasoning? To address this, the paper proposed an RL framework designed for distinguishing between perception and reasoning errors. The proposed method achieves a significant improvement.

All four reviewers provide consistent positive assessments after rebuttal (Accept x2, Weak Accept x2). They agree that this work provides valuable insights into perception-reasoning synergy, along with an elegant solution. The extensive experiments and ablations support the claims.

Reviewers raised concerns regarding the training-time cost, the limited evaluation on the Qwen backbone, the reliance on a stronger external text-only model, and the presentation quality. The authors provided a convincing rebuttal, and the reviewers generally found that the majority of concerns were resolved.

Overall, there is a clear consensus that the paper meets the bar for acceptance. Hence, I recommend accepting this paper.